# Wearable Technology and Its Influence on Motor Development and Biomechanical Analysis

**DOI:** 10.3390/ijerph21091126

**Published:** 2024-08-26

**Authors:** Pedro Morouço

**Affiliations:** 1ESECS, Polytechnic University of Leiria, 2411-901 Leiria, Portugal; pedro.morouco@ipleiria.pt; Tel.: +351-244-829-400; 2CIDESD, Research Center in Sports Sciences, Health Sciences and Human Development, 6201-001 Covilhã, Portugal

**Keywords:** wearable technology, motor development, biomechanics, artificial intelligence, monitoring, innovation

## Abstract

The convergence among biomechanics, motor development, and wearable technology redefines our understanding of human movement. These technologies allow for the continuous monitoring of motor development and the state of motor abilities from infancy to old age, enabling early and personalized interventions to promote healthy motor skills. For athletes, they offer valuable insights to optimize technique and prevent injuries, while in old age, they help maintain mobility and prevent falls. Integration with artificial intelligence further extends these capabilities, enabling sophisticated data analysis. Wearable technology is transforming the way we approach motor development and maintenance of motor skills, offering unprecedented possibilities for improving health, performance, and quality of life at every stage of life. The promising future of these technologies paves the way for an era of more personalized and effective healthcare, driven by innovation and interdisciplinary collaboration.

## 1. Introduction

Wearable electronic devices (also known as wearables) are innovative technologies designed to be worn on the body, offering a range of functionality for monitoring and analyzing biometric and movement data [1]. These devices include advanced sensors such as accelerometers, gyroscopes, and magnetometers, which capture detailed information about the body’s dynamics in real time [2]. Common examples of wearables are smart watches, fitness trackers, and smart clothing integrated with sensors. They can monitor heart rate, blood oxygen levels, sleep patterns, physical activity, and body posture [1]. Furthermore, these devices provide immediate feedback to the user, allowing for quick and accurate adjustments to daily activities or workout programs [2,3,4]. In the field of biomechanics, wearables have revolutionized the analysis of human movement, offering granular data that help optimize sports performance, prevent injuries, and improve physical rehabilitation [5,6,7,8]. In healthcare, they play a crucial role in continuously monitoring chronic conditions and promoting a healthy lifestyle [9]. The ability to collect and analyze biometric data non-invasively and continuously makes wearable devices a powerful tool for improving quality of life and physical performance in diverse populations [6].

Wearable technologies have revolutionized the way we understand and improve human movement by offering detailed, personalized insights that were previously inaccessible [1]. Equipped with precise sensors, these devices provide a comprehensive view of movement patterns. This continuous data collection allows for in-depth analysis of daily activities and specific exercises, helping to identify inefficiencies and potential injury risks [10]. For example, in athletes, wearables can monitor running technique, detect deviations in form, and provide immediate feedback for corrective adjustments [11]. Furthermore, in rehabilitation programs, these devices allow for more accurate and personalized monitoring of the patient’s progress, adjusting exercises and therapies based on objective and real-time data [12]. For instance, devices that monitor posture can alert the user to poor postural habits and suggest corrections, preventing long-term musculoskeletal problems [13]. In the elderly, wearables can detect falls and monitor mobility, facilitating quick and effective intervention [14,15].

Additionally, the integration of these technologies with mobile apps and artificial intelligence platforms allows for even more sophisticated analysis of the data collected. Machine learning algorithms can identify patterns and trends that are not immediately evident, providing a deeper understanding of human movement and making it easier to personalize interventions. Wearable technologies offer new ways to monitor, analyze, and improve human movement, promoting a more accurate and personalized approach to physical performance and health.

## 2. Wearable Technology

Types of wearable technology include motion sensors, smart clothing, and wearable devices [1]. Motion sensors, such as accelerometers and gyroscopes, monitor body dynamics in real time. Smart clothing embeds sensors into the fabrics, capturing physiological and movement data. Wearable devices, such as smart watches and fitness trackers, monitor physical activity, heart rate, and sleep patterns. Exoskeletons, another category, help with rehabilitation and improved mobility. These wearable technologies offer a diverse range of functionalities, enhancing the analysis and optimization of human movement and overall health [16].

Collecting accurate data during physical activity is critical to understanding and improving human performance [17]. Wearable technologies (e.g., accelerometers, gyroscopes, heart rate sensors) can capture detailed information about every movement in real time. These devices can monitor speed, acceleration, posture, and exercise intensity, providing a comprehensive analysis of the user’s biomechanics [5,7]. Accurate data allow the identification of inefficient movement patterns and injury risks, facilitating immediate, personalized adjustments. Also, continuous data collection offers valuable insights into workout progression, helping to optimize exercise routines and improve overall performance, whether for elite athletes or individuals looking to improve their health.

### 2.1. For Motor Development

Wearable technologies have a significant impact on motor development, from childhood to adulthood and aging. In children, wearable sensors can monitor motor development milestones such as crawling, walking, and running, providing accurate data on coordination and balance [18,19]. This allows for the early detection of anomalies and developmental delays, enabling personalized and effective interventions [20]. Children with conditions such as cerebral palsy can benefit from exoskeletons and smart clothing that aid in muscle rehabilitation and strengthening, promoting improvements in mobility and quality of life [21,22].

Overall, wearable technologies can have a profound impact on motor development, offering advanced tools to monitor, analyze, and improve movement at all stages of life. They promote a more personalized and precise approach to health and physical performance, benefiting everyone from children to the elderly.

#### Motor Development in Early Ages

Monitoring motor development during childhood and adolescence is crucial to ensure that children reach important milestones and develop fundamental motor skills in a healthy way [23]. Wearable technologies can play a vital role in this process, offering a non-invasive and seamless way to track motor progress and detect any abnormalities early. Sensors embedded in devices such as wristbands, smart clothing, and children’s exoskeletons can measure a variety of biomechanical parameters, including speed of movement, range of motion, posture, and balance [24,25,26].

In early childhood, these devices can monitor basic activities such as crawling, walking, and running. Accelerometers and gyroscopes built into smart clothing can track a baby’s movements, providing detailed data on coordination and balance. These data are crucial for identifying deviations from expected development milestones. Abnormalities detected early, such as delayed gait onset or balance difficulties, can be indicative of neuromotor conditions such as cerebral palsy [22,27]. Early detection allows parents and healthcare professionals to quickly intervene with specific therapies and exercises to correct or mitigate these problems [6].

During adolescence, the body undergoes several significant changes due to rapid growth and physical maturation. Wearable technologies can continue to monitor motor development, adapting to changing biomechanical needs. For instance, sensors can track body alignment and weight distribution during physical activities, helping to identify inappropriate movement patterns that could lead to injury [28]. This is especially important for young athletes, who are at risk of developing repetitive injuries due to strain or improper training techniques [29]. Moreover, continuous monitoring allows physical educators and trainers to tailor exercise programs to the individual needs of adolescents. Data collected on flexibility, strength, and endurance can be used to create personalized workout routines that promote balanced and healthy motor development [30]. For adolescents with chronic conditions or disabilities, wearable technologies can provide essential data to adjust therapeutic interventions and track progress in a more accurate and detailed way.

The integration of these technologies with digital platforms and mobile apps also facilitates data sharing between parents, educators, and healthcare providers [31]. This creates a collaborative approach to motor development monitoring, where all stakeholders can access up-to-date information and make informed decisions about the child’s or adolescent’s well-being. In addition, analyzing large volumes of data can help identify trends and patterns in larger populations, contributing to research and improving motor development practices.

Wearable technologies offer a powerful tool for monitoring motor development during childhood and adolescence. They enable the early detection of abnormalities, facilitate rapid and personalized interventions, and promote healthy motor development through accurate data and continuous analysis. With the increasing adoption of these technologies, a future is expected in which monitoring and improving motor development will be more accessible and effective for all children and adolescents.

## 3. Transformative Impact

Wearable technology is revolutionizing biomechanical analysis and motor development, providing a deeper and more accurate understanding of human movement [32]. Equipped with advanced sensors (e.g., accelerometers, gyroscopes, magnetometers) these devices capture data in real time, offering a detailed analysis of movement patterns. This continuous monitoring capability allows the identification of inefficiencies and injury risks that previously went unnoticed [6].

The integration of these technologies with mobile apps and artificial intelligence platforms allows for even more sophisticated data analysis. Machine learning algorithms identify patterns and trends, offering a deeper understanding of human movement. Wearable technology is transforming biomechanical analysis and motor development, promoting a more personalized, accurate, and effective approach to health and physical performance at all ages.

### 3.1. Via Biomechanics

The analysis of biomechanical variables in a child’s motor development is crucial to understand their progression and identify potential areas of concern or intervention [33]. Variables such as joint range of motion, motor coordination, balance, muscle strength, and gait patterns offer insights into a child’s ability to perform basic and complex motor activities. Also, factors such as speed, agility, posture, fine and gross motor control, reaction time, and adaptation to the environment are important indicators of global development. Analyzing these variables in a comprehensive and integrated way allows for a more accurate assessment of the child’s motor skills, enabling targeted interventions to promote healthy and balanced development [34].

In detail, research may provide significant insights over some relevant biomechanical variables to analyze a child’s motor development:Joint range of motion: In key joints, such as the shoulders, hips, and knees, it is crucial for performing basic and complex motor activities.Motor coordination: The ability to coordinate movements from different parts of the body, such as hand–eye coordination or bilateral coordination, is essential for the development of fundamental motor skills.Balance and stability: The ability to maintain static and dynamic balance is important for performing motor tasks such as walking, running, and jumping without falling.Muscle strength: Especially in key muscle groups like the trunk and leg muscles, it influences a child’s ability to perform effective and sustained motor movements.Gait patterns: Gait patterns, such as stride symmetry, stride length, and posture while walking, can provide insights into motor development and identify potential abnormalities.Speed and agility: The ability to move quickly and nimbly is important for participation in sports and recreational activities, as well as for injury prevention.Posture and body alignment: Proper posture and correct body alignment are critical for efficient motor performance and the prevention of injury over time.Fine and gross motor control: Precise control over fine movements, such as the fingers and hands, and the ability to control larger movements, such as the arms and legs, are important indicators of motor development.Reaction time: The time it takes to initiate a movement in response to an external stimulus can indicate the efficiency of the child’s nervous system and motor control.Adaptation to the environment: A child’s ability to adapt to the environment around them, such as when going up and down stairs, avoiding obstacles, and adjusting movement as needed, is essential for autonomy and safety while performing daily motor activities.

Analyzing these biomechanical variables can provide a comprehensive understanding of a child’s motor development and help identify areas that need support or intervention to promote healthy and balanced development.

### 3.2. Via Artificial Intelligence

Technological advancements in the field of wearable technology are driving an era of unprecedented innovation, with the continued development of increasingly advanced sensors and their integration with artificial intelligence (AI) promising to radically transform biomechanical analysis and motor development [35]. More advanced sensors, with improved data capture and processing capabilities, are being developed to offer a deeper and more comprehensive understanding of human movement. These sensors can be embedded in a variety of wearable devices, from smart clothing [36] to bioelectronic implants [37], and are designed to provide more accurate and detailed measurements of biomechanical parameters such as movement, strength, and physiological activity.

The integration of these sensors with AI is one of the key drivers of the next revolution in biomechanical analysis and motor development [38]. Machine learning algorithms and neural networks can analyze large volumes of data generated by wearable sensors in real time [37], identifying complex patterns and providing valuable insights into human movement [39]. Thus, this intelligent processing capability will allow for the further personalization of interventions and training programs, tailoring them to each person’s individual needs accurately and effectively. Additionally, AI can predict performance trends [40], identify injury risks [7], and suggest movement optimization strategies [41], providing a more proactive and preventative approach to healthcare and physical performance. In the future, wearable sensors are expected to become even more sophisticated and integrated, with the development of new emerging technologies such as flexible biosensors and nanotechnology [42]. These advancements will enable more discrete and seamless data collection, eliminating the limitations of traditional devices and expanding the applications of wearable technology to a variety of areas, from health and fitness to virtual and augmented reality. Furthermore, the miniaturization and cost reduction of electronic components will enable greater accessibility and widespread adoption of wearable technology across all layers of society.

With the continued evolution of wearable technology and its integration with AI, the future is expected to be marked by a revolution in how we understand, monitor, and improve human movement. These technologies have the potential to drive significant advancements in athletic performance, physical rehabilitation, injury prevention, and the promotion of health and wellness at all stages of life. The future of wearable technology is exciting and promising, offering endless possibilities for enhancing our understanding of the human body and improving our quality of life in ways that previously seemed impossible.

For example, an AI-based system could be used to analyze videos of babies during their first attempts to crawl or walk. AI algorithms can identify specific patterns of movement, such as the speed, amplitude, and symmetry of movements, and compare them to typical motor development milestones [43]. Based on this analysis, the system can provide feedback to parents or healthcare professionals on the child’s motor progress, indicating whether there are any delays or abnormalities that may require intervention [44]. Additionally, AI can be used to develop personalized intervention programs for children with motor difficulties. Based on the data collected about the child’s movement and data analysis algorithms, the system can identify specific areas of weakness or challenge and suggest activities and exercises tailored to the child’s individual needs [18]. This allows for a more targeted and effective approach to improving motor development and overcoming any obstacles the child may face.

In summary, the application of artificial intelligence in the field of child motor development can offer a range of benefits, from the early detection of anomalies to the development of personalized interventions. These systems have the potential to significantly improve the quality of childcare and ensure that all children reach their full motor potential.

## 4. Training

Apart from the abovementioned features, we envision significant problems in terms of the training required for technicians who use these technologies [45]. Technological evolution often outpaces the speed of adequate training, resulting in a knowledge gap between the development of tools and the ability of health professionals to use them effectively [46]. Healthcare technicians and professionals need ongoing training to keep up with innovations and maximize the potential of wearable technologies and artificial intelligence [16]. This knowledge gap can lead to the underutilization or misuse of devices, compromising the accuracy of the data collected and the effectiveness of interventions based on those data. Additionally, the complexity of AI systems and algorithms can be intimidating for those without specific technical training, making it difficult to correctly interpret the results and implement appropriate measures [47]. Insufficient training can also affect the confidence of health technicians and professionals in the use of these technologies, potentially limiting their adoption and integration into daily clinical practice. To address this issue, it is essential to develop accessible, continuing education programs that include both theory and practice [45]. Workshops, online courses, certifications, and collaboration with technology developers can help reduce this knowledge gap.

In addition, training must include ethical and privacy aspects, ensuring that professionals are prepared to deal with the sensitive data collected by these technologies [48]. Thus, continuous training is essential to ensure that wearable technologies and artificial intelligence are used effectively, safely, and ethically, maximizing their benefits for children’s motor development and overall health [49].

## 5. Ethical Considerations

Ethical issues and personal data security are key aspects in the use of wearable technologies and artificial intelligence in monitoring children’s motor development. The collection and analysis of biometric and movement data involves highly sensitive information, including health, location, and daily activity data. This raises several ethical and safety concerns that need to be carefully addressed [48].

First, data privacy is a critical concern. Wearable devices collect a vast amount of personal data that need to be protected from unauthorized access. This necessitates the use of robust security protocols, such as data encryption, strong authentication, and strict access control measures [47]. Ensuring that data are stored and transmitted securely is essential to protecting the privacy of individuals. Informed consent is another crucial ethical issue. Users, or in the case of children, their parents or guardians, should be fully informed about the types of data that will be collected, how they will be used, and who will have access to them. It is critical to ensure that consent is voluntary and that users have the option to revoke their consent at any time. In addition, transparency in the use of data is essential. The developers and operators of these technologies must be transparent about the algorithms used and how data-driven decisions are made. This includes providing clear and accessible explanations of how the data are analyzed and interpreted by artificial intelligence [50].

Data accountability and governance are also important aspects. Organizations that collect and utilize biometric data should have clear data governance policies, including appointing data protection officers and implementing regular audits to ensure compliance with data protection regulations, such as GDPR in Europe.

Prevention of bias and discrimination is another ethical concern. Artificial intelligence algorithms can inadvertently perpetuate or amplify existing biases in the data, leading to unfair or discriminatory decisions. It is crucial that AI systems are continuously monitored and adjusted to minimize these risks, ensuring that all individuals are treated fairly and equitably [51].

Finally, equity in access to technology is an important ethical consideration. It is necessary to ensure that these technologies are available to all layers of society, regardless of socioeconomic factors. This includes not only affordability but also the availability of educational resources so that everyone can benefit equally from these innovations. In summary, addressing the ethical and security issues of personal data is essential to ensure that wearable technologies and artificial intelligence are used responsibly and beneficially, protecting the privacy of individuals and promoting trust in the use of these advanced tools.

## 6. Limitations

While wearable technologies and artificial intelligence offer numerous advantages in monitoring and analyzing motor development, they also have some limitations [52]. Firstly, the accuracy of the data collected can be affected by the quality of the sensors and the correct placement of the devices on the body, which can result in inaccurate or inconsistent measurements. In addition, interpreting the data requires sophisticated and well-trained algorithms, which may not always be accurate or adaptable to all individual variabilities. Another important limitation is accessibility and cost. High-tech devices can be expensive, which can limit access for low-income populations or in regions with fewer resources. Usability and acceptance are also challenges, as not all children or their parents can be comfortable with the continued use of wearable devices, especially if they are intrusive or uncomfortable.

On other hand, data privacy and security are critical concerns, as continuous monitoring generates a large amount of sensitive personal data that need to be protected from unauthorized access and potential breaches [53]. In addition, the integration of these devices with broader healthcare systems can be complex and may require adequate infrastructure and interoperability between different technologies and platforms. Finally, there is the issue of technological dependence. Over-reliance on wearable technologies can lead to a decrease in traditional clinical observation and human interaction, which are irreplaceable in the holistic assessment of motor development [54]. From our point of view, these limitations highlight the need for a careful balance between the use of advanced technologies and traditional approaches in monitoring and promoting child motor development.

## 7. Conclusions

The intersection between biomechanics and motor development reveals a fascinating scenario driven by technological innovation and the advancement of artificial intelligence. Wearable technologies have emerged as a transformative tool, offering unprecedented insight into human movement at every stage of life. From infancy to old age, these devices play a crucial role in monitoring motor development and in the early detection of abnormalities.

In childhood and adolescence, the continuous monitoring of motor development provides a more accurate and comprehensive understanding of children’s progress, allowing for early and personalized interventions when needed. Wearable sensors and AI-based analytics systems empower parents and healthcare providers to identify developmental delays and provide targeted support to promote healthy motor skills from a young age. As in any development process, there are several limitations and concerns to be considered. While enthusiasm is evident, researchers should pay attention to the limitations, ethics, and training issues.

Advancements in wearable technology and artificial intelligence are transforming how we understand and improve human movement. These innovations provide unprecedented opportunities to monitor motor development, optimize athletic performance, and promote health and well-being at all ages. The future of biomechanics and motor development is exciting, with endless possibilities for enhancing the quality of life and unlocking human potential through technology.

## Data Availability

Not applicable.

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
