# Peer review of "Wearable Technology and Its Influence on Motor Development and Biomechanical Analysis"

_ijerph, 2024, doi:10.3390/ijerph21091126_

Round 1

Reviewer 1 Report

Comments and Suggestions for Authors

This is an interesting article describing the main uses of technology, including AI, in the field of motor development analysis.

They predict an evolution in their use in terms of how the understanding and improvement of human movement will be transformed by these technologies.

I think that from a descriptive point of view it is very correct, but I also think that there should be some concrete examples of this technology with examples of applications and results based on previous studies with their references.

On the other hand, I think it would be very appropriate to describe the limitations of this technology, as well as the part of the training that the technicians who implement it and analyse the data must have, since the technological evolution is faster than the training to be able to take advantage of it and be efficient with it. Finally, I think it is very important to highlight a key aspect such as ethical issues in the implementation of AI and the protection of users' personal data, because nothing is said about them.

Comments on the Quality of English Language

In my opinion, the quality of the English is very good and the text is developed in a very understandable way. There are only minor issues that could be addressed, but I think the quality is good.

Author Response

REV#1

This is an interesting article describing the main uses of technology, including AI, in the field of motor development analysis.

Dear Reviewer, thank you very much for your feedback.

They predict an evolution in their use in terms of how the understanding and improvement of human movement will be transformed by these technologies.

Dear Reviewer, thank you very much for your feedback.

I think that from a descriptive point of view it is very correct, but I also think that there should be some concrete examples of this technology with examples of applications and results based on previous studies with their references.

Dear Reviewer, thank you very much for your feedback. We have included examples of applications throughout the manuscript. Indeed, having these examples makes it easier for the reader to understand the state-of-the-art.

On the other hand, I think it would be very appropriate to describe the limitations of this technology, as well as the part of the training that the technicians who implement it and analyse the data must have, since the technological evolution is faster than the training to be able to take advantage of it and be efficient with it. Finally, I think it is very important to highlight a key aspect such as ethical issues in the implementation of AI and the protection of users' personal data, because nothing is said about them.

Dear Reviewer, thank you very much for your feedback. We do agree that it is quite important to present the limitations and ethical issues. With your suggestions we do think that the quality of the manuscript is much better now.

Comments on the Quality of English Language

In my opinion, the quality of the English is very good and the text is developed in a very understandable way. There are only minor issues that could be addressed, but I think the quality is good.

Dear Reviewer, thank you very much for your feedback. We have made a deep revision throughout the manuscript.

Reviewer 2 Report

Comments and Suggestions for Authors

A review of the manuscript titled:

Wearable Technology and its Influence on Motor Development and Biomechanical Analysis

I express my general approval of the work of the authors of this manuscript, it is interesting and worth publishing, but it basically lacks balance in terms of the positive aspect of the use of a valuable technology and the circumstances limiting its use and the approximation of this aspect.

Specific comments:

Abstract:

Line 11 Maybe better: (...) development and state of motor abilities from infancy to old age, (...)

Lines 14-15 I suggest to remove words - personalized interventions - which has already been expressed in this section;

Lines 15-16 Maybe: (…) motor development and maintenance of motor skills (…).

1.      Introduction:

P 1, line 24 (…),also known as wearables,(…) – please insert to paratheses;

P1, lines 41-42 (…) sensors such as accelerometers, gyroscopes, and magnetometers, these devices capture real-time (…) - This statement is unnecessary because it is repeated from the previous paragraph.

Lines 43-47  - These following sentences have the same message and end with the same mention of injuries.

Line 50 Third paragraph and third statement “real time”, and also further part of manuscript.

Lines 30 and 52  - Two following paragraphs and two “sleep patterns”

According to the reviewer, it should also be mentioned that the health and fitness benefits mentioned by the authors resulting from the use of specially manufactured intelligent electronics and AI will occur in people who willingly use them and in those who cannot be eliminated technologically for various reasons.

2.      Wearable Technology

This entire section, apart from the additional dedication to elite athletes, is a nice but repeated introduction.

2.1. For Motor Development and 2.1.1. Motor Development in Early Ages

This part is written correctly, especially the description of application, but there are repetitions of previously used statements.

A mention of adopting new devices, perhaps creating beneficial habits and customs related to the use of technology, would diversify this neatly written section.

P3, line 98 If we're talking about sports and swimming, please read about the use of accelerometers in swimming:

Propulsive limb coordination and body acceleration in sprint breaststroke swimming.

3.      Transformative Impact

This part sounds like a manifesto with a one-sided view of only the advantages, using lofty terms - unprecedented, comprehensive, next revolution, sophisticated, augmented, exciting and promising.

And on the other hand, there are those who have the right distance from using this advanced electronics, why? That's why, it would be a welcome addition.

There is no mention of some sociological aspects and other applications, circumstances, some people use it because they want and see the advantages, such as myself, others do not use it because:... etc.

4. Conclusion

As throughout, there is plenty of upbeat praise, but it lacks balance.

Author Response

REV#2

Wearable Technology and its Influence on Motor Development and Biomechanical Analysis

I express my general approval of the work of the authors of this manuscript, it is interesting and worth publishing, but it basically lacks balance in terms of the positive aspect of the use of a valuable technology and the circumstances limiting its use and the approximation of this aspect.

Dear Reviewer, thank you very much for your feedback. We do agree with your suggestions. Reading it now it is clear that mentioning the circumstances limiting its use is missing. We have included it following your suggestion.

Specific comments:

Abstract:

Line 11 Maybe better: (...) development and state of motor abilities from infancy to old age, (...)

Changed as suggested.

Lines 14-15 I suggest to remove words - personalized interventions - which has already been expressed in this section;

Changed as suggested.

Lines 15-16 Maybe: (…) motor development and maintenance of motor skills (…).

Changed as suggested.

  1. Introduction:

P 1, line 24 (…),also known as wearables,(…) – please insert to paratheses;

Changed as suggested.

P1, lines 41-42 (…) sensors such as accelerometers, gyroscopes, and magnetometers, these devices capture real-time (…) - This statement is unnecessary because it is repeated from the previous paragraph.

Changed as suggested.

Lines 43-47  - These following sentences have the same message and end with the same mention of injuries.

Changed as suggested.

Line 50 Third paragraph and third statement “real time”, and also further part of manuscript.

Changed as suggested.

Lines 30 and 52  - Two following paragraphs and two “sleep patterns”

Changed as suggested.

According to the reviewer, it should also be mentioned that the health and fitness benefits mentioned by the authors resulting from the use of specially manufactured intelligent electronics and AI will occur in people who willingly use them and in those who cannot be eliminated technologically for various reasons.

Dear Reviewer, thank you very much for your feedback. We do agree with your suggestions. Reading it now it is clear that mentioning the circumstances limiting its use is missing. We have included it following your suggestion.

  1. Wearable Technology

This entire section, apart from the additional dedication to elite athletes, is a nice but repeated introduction.

Dear Reviewer thank you for your feedback. We do understand that this section is similar to the introduction, but from our point of view it gives a little bit more of detail for those who want to get some more examples on technology. Thus, if you agree we would prefer to keep it.

2.1. For Motor Development and 2.1.1. Motor Development in Early Ages

This part is written correctly, especially the description of application, but there are repetitions of previously used statements.

A mention of adopting new devices, perhaps creating beneficial habits and customs related to the use of technology, would diversify this neatly written section.

P3, line 98 If we're talking about sports and swimming, please read about the use of accelerometers in swimming:

Propulsive limb coordination and body acceleration in sprint breaststroke swimming.

Dear Reviewer thank you for your feedback. We fully agree that there was some unnecessary repetition. We have proceeded accordingly.

  1. Transformative Impact

This part sounds like a manifesto with a one-sided view of only the advantages, using lofty terms - unprecedented, comprehensive, next revolution, sophisticated, augmented, exciting and promising.

And on the other hand, there are those who have the right distance from using this advanced electronics, why? That's why, it would be a welcome addition.

There is no mention of some sociological aspects and other applications, circumstances, some people use it because they want and see the advantages, such as myself, others do not use it because:... etc.

Dear Reviewer, thank you very much for your feedback. We do agree with your suggestions. Reading it now it is clear that mentioning the circumstances limiting its use is missing. We have included it following your suggestion.

  1. Conclusion

As throughout, there is plenty of upbeat praise, but it lacks balance.

Dear Reviewer, thank you very much for your feedback. We do agree with your suggestions.